# Invert to Learn to Invert

**Patrick Putzky**
Amlab, University of Amsterdam (UvA), Amsterdam, The Netherlands
Max-Planck-Institute for Intelligent Systems (MPI-IS), Tübingen, Germany
patrick.putzky@googlemail.com

**Max Welling**
Amlab, University of Amsterdam (UvA), Amsterdam, The Netherlands
Canadian Institute for Advanced Research (CIFAR), Canada
welling.max@googlemail.com

## Abstract

Iterative *learning to infer* approaches have become popular solvers for inverse problems. However, their memory requirements during training grow linearly with model depth, limiting in practice model expressiveness. In this work, we propose an iterative inverse model with constant memory that relies on invertible networks to avoid storing intermediate activations. As a result, the proposed approach allows us to train models with 400 layers on 3D volumes in an MRI image reconstruction task. In experiments on a public data set, we demonstrate that these deeper, and thus more expressive, networks perform state-of-the-art image reconstruction.

## 1 Introduction

We consider the task of solving inverse problems. An inverse problem is described through a so called forward problem that models either a real world measurement process or an auxiliary prediction task. The forward problem can be written as

$$\mathbf{d} = \mathcal{A}(\mathbf{p}, \mathbf{n}) \tag{1}$$

where $\mathbf{d}$ is the measurable data, $\mathcal{A}$ is a (non-)linear forward operator that models the measurement process, $\mathbf{p}$ is an unobserved signal of interest, and $\mathbf{n}$ is observational noise. Solving the inverse problem is then a matter of finding an inverse model $\mathbf{p} = \mathcal{A}^{-1}(\mathbf{d})$. However, if the problem is ill-posed or the forward problem is non-linear, finding $\mathcal{A}^{-1}$ is a non-trivial task. Oftentimes, it is necessary to impose assumptions about signal $\mathbf{p}$ and to solve the task in an iterative fashion [1].

### 1.1 Learn to Invert

Many recent approached to solving inverse problems focus on models that *learn to invert* the forward problem by mimicking the behaviour of an iterative optimization algorithm. Here, we will refer to models of this type as "iterative inverse models". Most iterative inverse models can be described through a recursive update equation of the form

$$\mathbf{p}_{t+1}, \mathbf{s}_{t+1} = h_\phi(\mathcal{A}, \mathbf{d}, \mathbf{p}_t, \mathbf{s}_t) \tag{2}$$

where $h_\phi$ is a parametric function, $\mathbf{p}_t$ is an estimate of signal $\mathbf{p}$ and $\mathbf{s}_t$ is an auxiliary (memory) variable at iteration $t$, respectively. Because $h_\phi$ is an iterative model it is often interpreted as a recurrent neural network (RNN). The functional form of $h_\phi$ ultimately characterizes different approaches to iterative inverse models. Figure 1 (A) illustrates the iterative steps of such models.

Training of iterative inverse models is typically done via supervised learning. To generate training data, measurements $\mathbf{d}$ are simulated from ground truth observations $\mathbf{p}$ through a known forward

**(A)** General form of Iterative Inverse Models

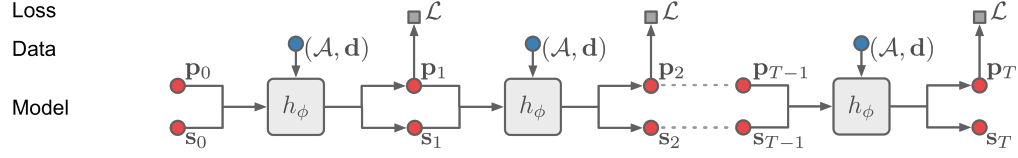

**(B)** RIM forward step

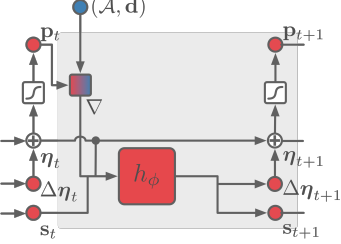

**(C)** i-RIM forward step

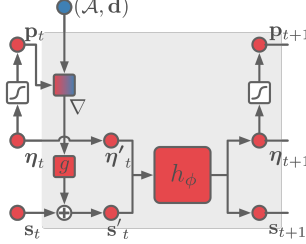

**(D)** i-RIM reverse step

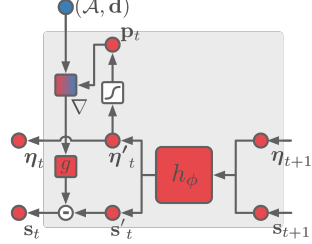

Figure 1: Iterative Inverse Models: Unrolled model and individual steps of RIM and i-RIM.

model $\mathcal{A}$. The training loss then takes the form

$$\mathcal{L}(\mathbf{p}; \mathbf{d}, \mathcal{A}, \phi) = \sum_t^T \omega_t \mathcal{L}\left(\mathbf{p}, \hat{\mathbf{p}}_t(\mathcal{A}, \mathbf{d}, \phi)\right) \tag{3}$$

where $\mathbf{p}$ is the ground truth signal, $\hat{\mathbf{p}}_t(\mathcal{A}, \mathbf{d}, \phi)$ is the model estimate at iteration $t$ given data $(\mathcal{A}, \mathbf{d})$ and model parameters $\phi$, and $\omega_t \in \mathbb{R}^+$ is an importance weight of the $t$-th iteration.

Models of this form have been successfully applied in several image restoration tasks on natural images [2–7], on sparse coding [8], and more recently have found adoption in medical image reconstruction [9–12], a field where sparse coding and compressed sensing have been a dominant force over the past years [1, 13].

## 1.2 Invert to Learn

In order to perform back-propagation efficiently, we are typically required to store intermediate forward activations in memory. This imposes a trade-off between model complexity and hardware memory constraints which essentially limits network depth. Since iterative inverse models are trained with back-propagation through time, they represent some of the deepest, most memory consuming models currently used. As a result, one often has to resort to very shallow models at each step of the iterative process. Here, we overcome these model limitations by presenting a memory efficient way to train very deep iterative inverse models. To do that, our approach follows the training principles presented in Gomez et al. [14]. To save memory, the authors suggested to use reversible neural network architectures which make it unnecessary to store intermediate activations as they can be restored from post-activations. Memory complexity in this approach is $\mathcal{O}(1)$ and computational complexity is $\mathcal{O}(L)$, where $L$ is the depth of the network. In the original approach, the authors utilized pooling layers which still required storing of intermediate activations at these layers. In practice, memory cost was hence not $\mathcal{O}(1)$. Briefly after, Jacobsen et al. [15] demonstrated that a fully invertible network inspired by RevNets [14] can perform as well as a non-invertible model on discriminative tasks, although the model was not trained with memory savings in mind. Here we will adapt invertible neural networks to allow for memory savings in the same way as in Gomez et al. [14]. We refer to this approach as "invertible learning".

## 1.3 Invertible Neural Networks

Invertible Neural Networks have become popular predominantly in likelihood-based generative models that make use of the change of variable formula:

$$p_{\mathbf{x}}(\mathbf{x}) = p_{\mathbf{y}}\left(f\left(\mathbf{x}\right)\right)\left|\det\left(\frac{\partial f(\mathbf{x})}{\partial \mathbf{x}^\top}\right)\right| \tag{4}$$

which holds for invertible $f(\mathbf{x})$ [16]. Under a fixed distribution $p_\mathbf{y}(\cdot)$, an invertible model $f(\mathbf{x})$ can be optimized to maximize the likelihood of data $\mathbf{x}$. Dinh et al. [16] suggested an architecture for invertible networks in which inputs and outputs of a layer are split into two parts such that $\mathbf{x} = (\mathbf{x}_1, \mathbf{x}_2)$ and $\mathbf{y} = (\mathbf{y}_1, \mathbf{y}_2)$. Their suggested invertible layer has the form

$$
\begin{aligned}
\mathbf{y}_1 &= \mathbf{x}_1 & \mathbf{x}_2 &= \mathbf{y}_2 - \mathcal{G}(\mathbf{y}_1) \\
\mathbf{y}_2 &= \mathbf{x}_2 + \mathcal{G}(\mathbf{y}_1) & \mathbf{x}_1 &= \mathbf{y}_1
\end{aligned}
\tag{5}
$$

where the left-hand equations reflect the forward computation and the right-hand equation are the backward computations. This layer is similar to the layer used later in Gomez et al. [14]. Later work on likelihood-based generative models built on the idea of invertible networks by focusing on non-volume preserving layers [17, 18]. A recent work avoids the splitting from Dinh et al. [16] and focuses instead on $f(\mathbf{x})$ that can be numerically inverted [19].

Here, we will assume that any invertible neural network can be used as a basis for invertible learning. Apart from the fact that our approach will allow us to have significant memory savings during training, it can further allow us to leverage Eq. (4) for unsupervised training in the future.

### 1.4 Recurrent Inference Machines

We will use Recurrent Inference Machines (RIM) [2] as a basis for our iterative inverse model. The update equations of the RIM take the form

$$
\mathbf{s}_{t+1} = f\left(\nabla\mathcal{D}\left(\mathbf{d}, \mathcal{A}\left(\Psi\left(\boldsymbol{\eta}_t\right)\right)\right), \boldsymbol{\eta}_t, \mathbf{s}_t\right)
\tag{6}
$$

$$
\boldsymbol{\eta}_{t+1} = \boldsymbol{\eta}_t + g\left(\nabla\mathcal{D}\left(\mathbf{d}, \mathcal{A}\left(\Psi\left(\boldsymbol{\eta}_t\right)\right)\right), \boldsymbol{\eta}_t, \mathbf{s}_{t+1}\right),
\tag{7}
$$

with $\mathbf{p}_t = \Psi(\boldsymbol{\eta}_t)$, where $\Psi$ is a link function, and $\mathcal{D}\left(\mathbf{d}, \mathcal{A}\left(\Psi\left(\boldsymbol{\eta}_t\right)\right)\right)$ is a data consistency term which ensures that estimates of $\mathbf{p}_t$ stay true to the measured data $\mathbf{d}$ under the forward model $\mathcal{A}$. Below, we will call $(\boldsymbol{\eta}_t, \mathbf{s}_t)$ the machine state at time t. The benefit of the RIM is that it simultaneously learns iterative inference and it implicitly learns a prior over $\mathbf{p}$. The update equations are general enough that they allow us to use any network for $h_\phi$. Unfortunately, even if $h_\phi$ was invertible, the RIM in it's current form is not. We will show later how a simple modification of these equations can make the whole iterative process invertible. An illustration of the update block for the machine state can be found in figure 1 (B).

### 1.5 Contribution

In this work, we marry iterative inverse models with the concept of invertible learning. This leads to the following contributions:

1. **The first iterative inverse model that is fully invertible.** This allows us to overcome memory constraints during training with invertible learning [14]. It will further allow for semi- and unsupervised training in the future [16].

2. **Stable invertible learning of very deep models.** In practice, invertible learning can be unstable due to numerical errors that accumulate in very deep networks [14]. In our experience, common invertible layers [16–18] suffered from this problem. We give intuitions why these layers might introduce training instabilities, and present a new layer that addresses these issues and enables stable invertible learning of very deep networks (400 layers).

3. **Scale to large observations.** We demonstrate in experiments that our model can be trained on large volumes in MRI (3d). Previous iterative inverse models were only able to perform reconstruction on 2d slices. For data that has been acquired with 3d sequences, however, these approaches are not feasible anymore. Our approach overcomes this issue. This result has implications in other domains with large observations such as synthesis imaging in radio astronomy.

## 2 Method

Our approach consists of two components which we will describe in the following section. (1) We present a simple way to modify Recurrent Inference Machines (RIM) [2] such that they become fully invertible. We call this model "invertible Recurrent Inference Machines" (i-RIM). (2) We present a new invertible layer with modified ideas from Kingma and Dhariwal [18] and Dinh et al. [16] that allows stable invertible learning of very deep i-RIMs. We briefly discuss why invertible learning with

conventional layers [16–18] can be unstable. An implementation of our approach can be found at `https://github.com/pputzky/invertible_rim`.

## 2.1 Invertible Recurrent Inference Machines

In the following, we will assume that we can construct an invertible neural network $h(\cdot)$ with memory complexity $\mathcal{O}(1)$ during training using the approach in Gomez et al. [14]. That is, if the network can be inverted layer-wise such that

$$h = h^L \circ h^{L-1} \circ \cdots \circ h^1 \tag{8}$$

$$h^{-1} = (h^1)^{-1} \circ (h^2)^{-1} \circ \cdots \circ (h^L)^{-1} \tag{9}$$

where $h^l$ is the $l$-th layer, and $(h^l)^{-1}$ is it's inverse, we can use invertible learning do back-propagation without storing activations in a computationally efficient way.

Even though we might have an invertible $h(\cdot)$, the RIM in the formulation given in Eq. (7) and Eq. (6) cannot trivially be made invertible. One issue is that if naively implemented, $h(\cdot)$ takes three inputs but has only two outputs as $\nabla \mathcal{D}(\mathbf{d}, \mathcal{A}\Psi \boldsymbol{\eta}_t)$ is not part of the machine state $(\boldsymbol{\eta}_t, \mathbf{s}_t)$. With this, $h(\cdot)$ would have to increase in size with the number of iterations of the model in order to stay fully invertible. The next issue is that the incremental update of $\boldsymbol{\eta}_t$ in Eq. (7) is conditioned on $\eta_t$ itself, which prevents us to use the trick from Eq. (5). One solution would be to save all intermediate $\eta_t$ but then we would still have memory complexity $\mathcal{O}(T) < \mathcal{O}(T * L)$ which is an improvement but still too restrictive for large scale data (compare 3.4, i-RIM 3D). Alternatively, Behrmann et al. [19] show that Eq. (7) could be numerically inverted if $\mathrm{Lip}(h) < 1$ in $\boldsymbol{\eta}$. In order to satisfy this condition, this would involve not only restricting $h(\cdot)$ but also on $\mathcal{A}$, $\Psi$, and $\mathcal{D}(\cdot, \cdot)$. Since we want our method to be usable with any type of inverse problem described in Eq. (1) such an approach becomes infeasible. Further, for very deep $h(\cdot)$ numerical inversion would put a high computational burden on the training procedure.

We are therefore looking for a way to make the update equations of the RIM trivially invertible. To do that we use the same trick as in Eq. (5). The update equations of our invertible Recurrent Inference Machines (i-RIM) take the form (left - forward step; right - reverse step):

$$
\begin{aligned}
\boldsymbol{\eta}_t' &= \boldsymbol{\eta}_t & \boldsymbol{\eta}_t', \mathbf{s}_t' &= h_t^{-1}(\boldsymbol{\eta}_{t+1}, \mathbf{s}_{t+1}) \\
\mathbf{s}_t' &= \mathbf{s}_t + g_t(\nabla \mathcal{D}(\mathbf{d}, \mathcal{A}(\Psi(\boldsymbol{\eta}_t')))) & \mathbf{s}_t &= \mathbf{s}_t' - g_t(\nabla \mathcal{D}(\mathbf{d}, \mathcal{A}(\Psi(\boldsymbol{\eta}_t')))) \\
\boldsymbol{\eta}_{t+1}, \mathbf{s}_{t+1} &= h_t(\boldsymbol{\eta}_t', \mathbf{s}_t') & \boldsymbol{\eta}_t &= \boldsymbol{\eta}_t'
\end{aligned}
\tag{10}
$$

where we do not require weight sharing over iterations $t$ in $g_t$ or $h_t$, respectively. As can be seen, the update from $(\boldsymbol{\eta}_t, \mathbf{s}_t)$ to $(\boldsymbol{\eta}_t', \mathbf{s}_t')$ is reminiscent of Eq. (5), we assume that $h(\cdot)$ is an invertible function. Given $h(\cdot)$ can be inverted layer-wise as above, we can train an i-RIM model using invertible learning with memory complexity $\mathcal{O}(1)$. We have visualised the forward and reverse updates of the machine state in an i-RIM in figure 1(C) & (D).

## 2.2 An Invertible Layer with Orthogonal 1x1 convolutions

Here, we introduce a new invertible layer that we will use to form $h(\cdot)$. Much of the recent work on invertible neural networks has focused on modifying the invertible layer of Dinh et al. [16] in order to improve generative modeling. Dinh et al. [17] proposed a non-volume preserving layer that can still be easily inverted:

$$
\begin{aligned}
\mathbf{y}_1 &= \mathbf{x}_1 & \mathbf{x}_2 &= (\mathbf{y}_2 - \mathcal{G}(\mathbf{y}_1)) \odot \exp(-\mathcal{F}(\mathbf{y}_1)) \\
\mathbf{y}_2 &= \mathbf{x}_2 \odot \exp(\mathcal{F}(\mathbf{y}_1)) + \mathcal{G}(\mathbf{y}_1) & \mathbf{x}_1 &= \mathbf{y}_1
\end{aligned}
\tag{11}
$$

While improving over the volume preserving layer in Dinh et al. [16], the method still required manual splitting of layer activations using a hand-chosen mask. Kingma and Dhariwal [18] addressed this issue by introducing invertible $1 \times 1$ convolutions to embed the activations before using the affine layer from Dinh et al. [17]. The idea behind this approach is that this convolution can learn to permute the signal across the channel dimension to make the splitting a parametric approach. Following this, Hoogeboom et al. [20] introduced a more general form of invertible convolutions which operate not only on the channel dimension but also on spatial dimensions.

We have tried to use the invertible layer of Kingma and Dhariwal [18] in our i-RIM but without success. We suspect three possible causes of this issue which we will use as motivation for our proposed invertible layer:

**(A)** Invertible Layer

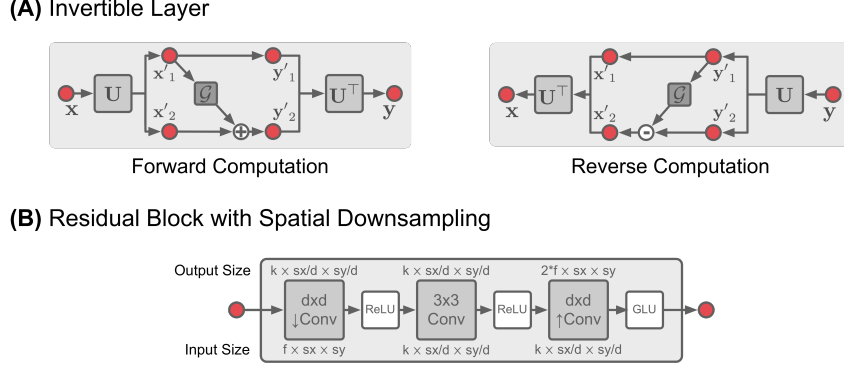

Forward Computation                             Reverse Computation

**(B)** Residual Block with Spatial Downsampling

Figure 2: Illustration of the layer used in this work. (A): Invertible layer with orthogonal $1 \times 1$ convolutional embedding. (B): Function $\mathcal{G}$ used in each invertible layer.

1. **Channel permutations** Channel ordering has a semantic meaning in the i-RIM (see Eq. (**??**)) which it does not have a priori in the above described likelihood-based generative models. Further, a permutation in layer $l$ will affect all channel orderings in down-stream layers $k > l$. Both factors may harm the error landscape.

2. **Exponential gate** The rescaling term $\exp(\mathcal{F}(\mathbf{x}))$ in Eq. (11) can make inversions numerically unstable if not properly taken care of.

3. **Invertible 1x1 convolutions** Without any restrictions on the eigenvalues of the convolution it is possible that it will cause vanishing or exploding gradients during training with back-propagation.

For the of the i-RIM, we have found an invertible layer that addresses all the potential issues mentioned above, is simple to implement, and leads to good performance as can be seen later. Our invertible layer has the following computational steps:

$$\mathbf{x}' = \mathbf{U}\mathbf{x} \tag{12}$$

$$\mathbf{y}'_1 = \mathbf{x}'_1 \tag{13}$$

$$\mathbf{y}'_2 = \mathbf{x}'_2 + \mathcal{G}(\mathbf{x}'_1) \tag{14}$$

$$\mathbf{y} = \mathbf{U}^\top \mathbf{y}' \tag{15}$$

where $\mathbf{x}' = (\mathbf{x}'_1, \mathbf{x}'_2)$ and $\mathbf{y}' = (\mathbf{y}'_1, \mathbf{y}'_2)$, and $\mathbf{U}$ is an orthogonal $1 \times 1$ convolution which is key to our invertible layer. Recall, the motivation for an invertible $1 \times 1$ convolution in Kingma and Dhariwal [18] was to learn a parametric permutation over channels in order to have a suitable embedding for the following affine transformation from Dinh et al. [16, 17]. An orthogonal $1 \times 1$ convolution is sufficient to implement this kind of permutation but will not cause any vanishing or exploding gradients during back-propagation since all eigenvalues of the matrix are 1. Further, $\mathbf{U}$ can be trivially inverted with $\mathbf{U}^{-1} = \mathbf{U}^\top$ which will reduce computational cost in our training procedure as we require layer inversions at every optimization step. Below, we will show how to construct an orthogonal $1 \times 1$ convolution. Another feature of our invertible layer is Eq. (15) in which we project the outputs of the affine layer back to the original basis of $\mathbf{x}$ using $\mathbf{U}^\top$. This means that our $1 \times 1$ convolution will act only locally on it's layer while undoing the permutation for downstream layers. A schematic of our layer and it's inverse can be found in figure 2.Here, we will omit the inversion of our layer for brevity and refer the reader to the supplement.

### 2.2.1 Orthogonal 1x1 convolution

A $1 \times 1$ convolution can be implemented through using a $k \times k$ matrix that is reshaped into a convolutional filter and then used in a convolution operation [18]. In order to guarantee that this matrix is orthogonal we use the method utilized in Tomczak and Welling [21] and Hoogeboom et al. [20]. Any orthogonal $k \times k$ matrix $\mathbf{U}$ can be constructed from a series of Householder reflections such that

$$\mathbf{U} = \mathbf{H}_K \mathbf{H}_{K-1} \dots \mathbf{H}_1 \tag{16}$$

where $\mathbf{H}_k$ is a Householder reflection with $\mathbf{H}_k = \mathbf{I} - 2\frac{\mathbf{v}_k \mathbf{v}_k^\top}{\|\mathbf{v}_k\|^2}$, and $\mathbf{v}_k$ is a vector which is orthogonal to the reflection hyperplane. In our approach, the vectors $\mathbf{v}_k$ represent the parameters to be optimized.

Because the Householder matrix is itself orthogonal, a matrix $\mathbf{U}_D$ that is constructed from a subset $D < K$ of Householder reflections such that $\mathbf{U}_D = \mathbf{H}_D \mathbf{H}_{D-1} \ldots \mathbf{H}_1$ is still orthogonal. We will use this fact to reduce the amount of parameters necessary to train on the one hand and to reduce the cost of constructing $\mathbf{U}_D$ on the other. For our experiments we chose $D = 3$.

### 2.2.2 Residual Block with Spatial Downsampling

In this work we utilize a residual block that is inspired by the multiscale approach used in Jacobsen et al. [15]. However, instead of shuffling pixels in the machine state, we perform this action in the residual block. Each block consists of three convolutional layers. The first layer performs a spatial downsampling operation of factor $d$, i.e. the convolutional filter will have size $d$ in every dimension, and we perform a strided convolution with stride $d$. This is equivalent to shuffeling pixels in a $d \times d$ patch into different channels of the same pixel followed by a $1 \times 1$ convolution. The second convolutional layer is a simple $3 \times 3$ convolution with stride 1. And the last convolutional layer reverses the spatial downsampling operation with a transpose convolution. At the output we have found that a Gated Linear Unit [22] guarantees stable training without the need for any special weight initializations. We use weight normalisation [23] for all convolutional weights in the block and we disable the bias term for the last convolution. Our residual block has two parameters: $d$ for the spatial downsampling factor, and $k$ for the number of channels in the hidden layers. An illustration of our residual block can be found in figure 2.

### 2.3 Related Work

Recently, Ardizzone et al. [24] proposed another approach of modeling inverse problems with invertible networks. In their work, however, the authors assume that the forward problem is unknown, and possibly as difficult as the inverse problem. In our case the forward problem is typically much easier to solve than the inverse problem. The authors suggest to train the network bi-directionally which could potentially help our approach as well. The RIM has found successful applications to several imaging problems in MRI [25, 9] and radio astronomy [26, 27]. We expect that our presented results can be translated to other applications of the RIM as well.

## 3 Experiments

We evaluate our approach on a public data set for accelerated MRI reconstruction that is part of the so called fastMRI challenge [28]. Comparisons are made between the U-Net baseline from Zbontar et al. [28], an RIM [2, 9], and an i-RIM, all operating on single 2D slices. To explore future directions and push the memory benefits of our approach to the limit we also trained a 3D i-RIM. An improvement on the results presented below can be found in Putzky et al. [29].

### 3.1 Accelerated MRI

The problem in accelerated Magnetic Resonance Imaging (MRI) can be described as a linear measurement problem of the form

$$\mathbf{d} = \mathbf{P}\mathscr{F}\boldsymbol{\eta} + \mathbf{n} \tag{17}$$

where $\mathbf{P} \in \mathbb{R}^{m \times n}$ is a sub-sampling matrix, $\mathscr{F}$ is a Fourier transform, $\mathbf{d} \in \mathbb{C}^m$ is the sub-sampled K-space data, and $\boldsymbol{\eta} \in \mathbb{C}^n$ is an image. Here, we assume that the data has been measured on a single coil, hence the measurement equation leaves out coil sensitivity models. This corresponds to the 'Single-coil task' in the fastMRI challenge [28]. Further, we set $\Psi$ to be the identity function.

### 3.2 Data

All of our experiments were run on the single-coil data from Zbontar et al. [28]. The data set consists of 973 volumes or 34,742 slices in the training set, 199 volumes or 7,135 slices in the validation set, and 108 volumes or 3,903 slices in the test set. While training and validation sets are both fully sampled, the test set is only sub-sampled and performance has to be evaluated through the fastMRI website [1]. All volumes in the data set have vastly different sizes. For mini-batch training we therefore reduced the size of image slices to $480 \times 320$ (2D models) and volume size to $32 \times 480 \times 320$

Table 1: Comparison of memory consumption during training and testing.

|  | RIM | i-RIM 2D | i-RIM 3D |
|---|---|---|---|
| Size Machine State $(\boldsymbol{\eta}, \mathbf{s})$ (CDHW) | $130 \times 1 \times 480 \times 320$ | $64 \times 1 \times 480 \times 320$ | $64 \times 32 \times 480 \times 320$ |
| Memory Machine State $(\boldsymbol{\eta}, \mathbf{s})$ (in GB) | 0.079 | 0.039 | 1.258 |
| Number of steps $T$ | 1/4/8 | 1/4/8 | 1/4/8 |
| Network Depth (#layers) | 5/20/40 | 50/200/400 | 50/200/400 |
| Memory during Testing (in GB) | 0.60 / 0.65 / 0.65 | 0.20 / 0.24 / 0.31 | 5.87/6.03 / 6.25 |
| Memory during Training (in GB) | 2.65 / 6.01 / 10.49 | 2.47 / 2.49 / 2.51 | 11.51 / 11.76 / 11.89 |

Table 2: Reconstruction performance on validation and test data from the fastMRI challenge [28] under different metrics. NMSE - normalized mean-squared-error (lower is better); PSNR - peak signal-to-noise ratio (higher is better); SSIM - structural similarity index [30] (higher is better).

|  | 4x Acceleration | | | 8x Acceleration | | |
|---|---|---|---|---|---|---|
| **Validation** | NMSE | PSNR | SSIM | NMSE | PSNR | SSIM |
| U-Net [28] | 0.0342 | 31.91 | 0.722 | 0.0482 | 29.98 | 0.656 |
| RIM | 0.0332 | 32.24 | 0.725 | 0.0484 | 30.03 | 0.656 |
| i-RIM 2D | **0.0316** | **32.55** | **0.734** | **0.0429** | **30.76** | **0.669** |
| i-RIM 3D | 0.0322 | 32.39 | 0.731 | 0.0435 | 30.66 | 0.667 |
| **Test** | NMSE | PSNR | SSIM | NMSE | PSNR | SSIM |
| U-Net [28] | 0.0320 | 32.22 | 0.754 | 0.0480 | 29.45 | 0.651 |
| RIM | 0.0270 | 33.39 | 0.759 | 0.0458 | 29.71 | 0.650 |
| i-RIM 2D | **0.0255** | **33.72** | **0.767** | **0.0408** | **30.41** | **0.664** |
| i-RIM 3D | 0.0261 | 33.54 | 0.764 | 0.0413 | 30.34 | 0.662 |

(3d model). Not all training volumes had 32 slices and hence were excluded for training the 3D model. During validation and test, we did not reduce the size of slices and volumes for reconstruction. During training, we simulated sub-sampled K-space data using Eq. (17). As sub-sampling masks we used the masks from the test set (108 distinct masks) in order to simulate the corruption process in the test set. For validation, we generated sub-sampling masks in the way described in Zbontar et al. [28]. Performance was evaluated on the central $320 \times 320$ portion of each image slice on magnitude images as in Zbontar et al. [28]. For evaluation, all slices were treated independently.

### 3.3 Training

For the Unet, we followed the training protocol from Zbontar et al. [28]. All other models were trained to reconstruct a complex valued signal. Real and imaginary parts were treated as separate channels as done in Lønning et al. [25]. The machine state $(\boldsymbol{\eta}_0, \mathbf{s}_0)$ was initialized with zeros for all RIM and i-RIM models. As training loss, we chose the normalized mean squared error (NMSE) which showed overall best performance. To regularize training we masked model estimate $\hat{\mathbf{x}}$ and target $\mathbf{x}$ with the same random sub-sampling mask $\mathbf{m}$ such that

$$\mathcal{L}^*(\hat{\mathbf{x}}) = NMSE(\mathbf{m} \odot \hat{\mathbf{x}}, \mathbf{m} \odot \mathbf{x}) \tag{18}$$

This is a simple way to add gradient noise in cases where the number of pixels is very large. We chose a sub-sampling factor of $0.01$, i.e. on average $1\%$ of pixels (voxels) from each sample were used during a back-propagation step. All iterative models were trained on $8$ inference steps.

For the RIM we used a similar model as in Lønning et al. [25]. The model consists of three convolutional layers and two gated recurrent units (GRU) [31] with 64 hidden channels each. During training the loss was averaged across all times steps. For the i-RIM, we chose similar architectures for the 2D and 3D models, respectively. The models consisted of 10 invertible layers with a fanned downsampling structure at each time step, no parameter sharing was applied across time steps, the loss was only evaluated on the last time step. The only difference between 2D and 3D model was that the former used 2D convolutions and the latter used 3d convolutions. More details on the model architectures can be found in Appendix B. As a model for $g_t$ we chose for simplicity

$$g_t(\nabla \mathcal{D}(\mathbf{d}, \mathcal{A}(\Psi(\boldsymbol{\eta}'_t)))) = \begin{pmatrix} \nabla \mathcal{D}(\mathbf{d}, \mathcal{A}(\Psi(\boldsymbol{\eta}'_t))) \\ \mathbf{0}_{D-2} \end{pmatrix} \tag{19}$$

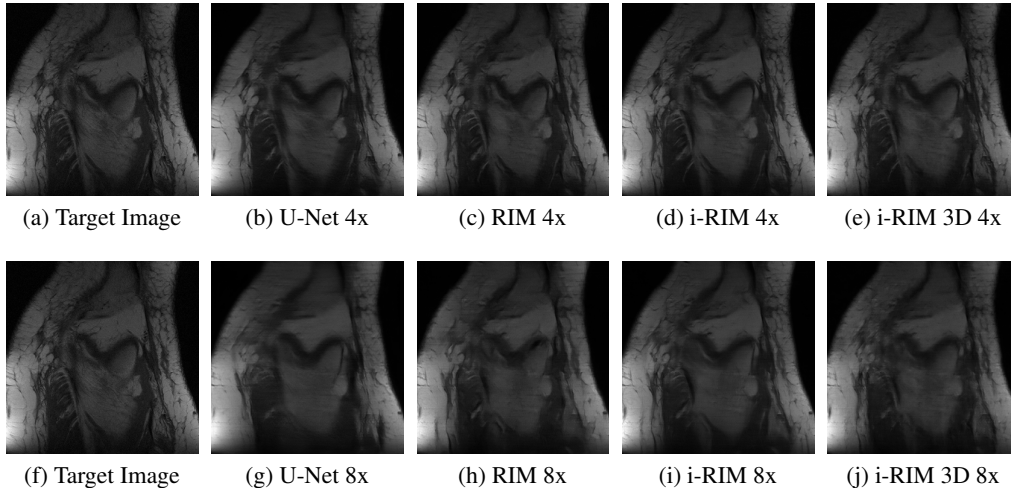

|                |                |             |               |                 |
|----------------|----------------|-------------|---------------|-----------------|
| (a) Target Image | (b) U-Net 4x | (c) RIM 4x | (d) i-RIM 4x | (e) i-RIM 3D 4x |

| (f) Target Image | (g) U-Net 8x | (h) RIM 8x | (i) i-RIM 8x | (j) i-RIM 3D 8x |

Figure 3: Reconstructions of a central slice in volume 'file1001458.h5' from the validation set. Top: 4x acceleration. Bottom: 8x acceleration. Zoom in for better viewing experience.

where $\mathbf{0}_{D-2}$ is a zero-filled tensor with $D-2$ number of channels with $D$ the number of channels in $\mathbf{s}_t$, which is the simplest model for $g$ we could use. The gradient information will be mixed in the downstream processing of $h$. We chose the number of channels in the machines state $(\boldsymbol{\eta}, \mathbf{s})$ to be $64$.

### 3.4 Results

The motivation of our work was to reduce memory consumption of iterative inverse models. In order to emphasize the memory savings achieved with our approach we compare memory consumption of each model in table 1. Shown are data for the machine state, and memory consumption during training and testing on a single GPU. As can be seen, for the 3D model we have a machine state that occupies more that $1.25$GB of memory ($7.5\%$ of available memory in a $16$GB GPU). It would have been impossible to train an RIM with this machine state given current hardware. Also note that memory consumption is mostly independent of network depth for both i-RIM models. A small increase in memory consumption is due to the increase of the number of parameters in a deeper model. We have thus introduced a model for which network depth becomes mostly a computational consideration.

We compared all models on the validation and test set using the metrics suggested in Zbontar et al. [28]. A summary of this comparison can be found in table 2. Both i-RIM models consistently outperform the baselines. At the time of this writing, all three models sit on top of the challenge's Single-Coil leaderboard.[2] The i-RIM 3D shows almost as good performance as it's 2D counterpart and we believe that with more engineering and longer training it has the potential to outperform the 2D model. A qualitative assessment of reconstructions of a single slice can be found in figure 3. We chose this slice because it contains a lot of details which emphasize the differences across models.

## 4   Discussion

We proposed a new approach to address the memory issues of training iterative inverse models using invertible neural networks. This enabled us to train very deep models on a large scale imaging task which would have been impossible to do with earlier approaches. The resulting models learn to do state-of-the-art image reconstruction. We further introduced a new invertible layer that allows us to train our deep models in a stable way. Due to it's structure our proposed layer lends itself to structured prediction tasks, and we expect it to be useful in other such tasks as well. Because invertible neural networks have been predominantly used for unsupervised training, our approach naturally allows us to exploit these directions as well. In the future, we aim to train our models in an unsupervised or semi-supervised fashion.

**Acknowledgements**

The authors are grateful for helpful comments from Matthan Caan, Clemens Korndörfer, Jörn Jacobsen, Mijung Park, Nicola Pezzotti, and Isabel Valera.

Patrick Putzky is supported by the Netherlands Organisation for Scientific Research (NWO) and the Netherlands Institute for Radio Astronomy (ASTRON) through the big bang, big data grant.

## Footnotes

[1] http://fastmri.org

[2]`http://fastmri.org/leaderboards`, Team Name: NeurIPS_Anon; Model aliases: RIM - model_a, i-RIM 2D - model_b, i-RIM 3D - model_c. See Supplement for screenshot.

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
