[Supplementary Material · NeurIPS_2019_Supplement__Invert_to_Learn_to_Invert.pdf]

# Supplement to: Invert to Learn to Invert

## A Reverse Computation of Invertible Layer

Figure 1: Illustration of our invertible layer. a)-d): Computational steps of the invertible layer. a) data $\mathbf{x}$; b) $\mathbf{x}' = \mathbf{U}\mathbf{x}$; c) $\mathbf{y}'$; d) $\mathbf{y} = \mathbf{U}^\top \mathbf{y}'$; e) Latent space $\mathbf{y}$ after unsupervised maximum likelihood training of a single invertible layer on data from a).

For completeness, we repeat here the computational steps of our invertible layer:

$$\mathbf{x}' = \mathbf{U}\mathbf{x} \tag{1}$$

$$\mathbf{y}'_1 = \mathbf{x}'_1 \tag{2}$$

$$\mathbf{y}'_2 = \mathbf{x}'_2 + \mathcal{G}(\mathbf{x}'_1) \tag{3}$$

$$\mathbf{y} = \mathbf{U}^\top \mathbf{y}' \tag{4}$$

where $\mathbf{x}' = (\mathbf{x}'_1, \mathbf{x}'_2)$ and $\mathbf{y}' = (\mathbf{y}'_1, \mathbf{y}'_2)$, $\mathbf{U}$ is an orthogonal $1 \times 1$ convolution, and $\mathcal{G}$ is any (non-)linear function.

The reverse computation then takes the form:

$$\mathbf{y}' = \mathbf{U}\mathbf{y} \tag{5}$$

$$\mathbf{x}'_2 = \mathbf{y}'_2 - \mathcal{G}(\mathbf{y}'_1) \tag{6}$$

$$\mathbf{x}'_1 = \mathbf{y}'_1 \tag{7}$$

$$\mathbf{x} = \mathbf{U}^\mathbf{T} \mathbf{x}' \tag{8}$$

And the total derivatives of intermediate layers are given by

$$\bar{\mathbf{y}}' = \mathbf{U}^\top \bar{\mathbf{y}} \tag{9}$$

$$\bar{\mathbf{x}}'_2 = \bar{\mathbf{y}}'_2 \tag{10}$$

$$\bar{\mathbf{x}}'_1 = \bar{\mathbf{y}}'_1 + \left(\frac{\partial \mathcal{G}}{\partial \mathbf{y}'_1}\right)^\top \bar{\mathbf{y}}'_2 \tag{11}$$

$$\bar{\mathbf{x}} = \mathbf{U}\bar{\mathbf{x}}' \tag{12}$$

where $\bar{\mathbf{x}}' = (\bar{\mathbf{x}}'_1, \bar{\mathbf{x}}'_2)$ and $\bar{\mathbf{y}}' = (\bar{\mathbf{y}}'_1, \bar{\mathbf{y}}'_2)$, and we use the same notation as in Gomez et al. [1] for the total derivative.

## B Details on model architectures and training

For the RIM we used a similar model as in Lønning et al. [2]. The model consists of three convolutional layers and two gated recurrent units (GRU) [3] with $64$ hidden channels each. The structure is (Conv2D(kernel=$5 \times 5$), GRU(kernel=$1 \times 1$),Conv2D(kernel=$3 \times 3$,dilation=2),GRU(kernel=$1 \times 1$), Conv2D(kernel=$3 \times 3$, bias=False)), with output size of 2 channels for the real and imaginary component, respectively. During training the loss was averaged across all times steps.

For the i-RIM, we chose similar architectures for the 2D and 3D models, respectively. The models consist of $10$ invertible layers at each time step, no parameter sharing was done across time steps, the loss was only evaluated on the last time step. The downsampling blocks at each time step were (d=(1,1,2,2,4,4,2,2,1,1),k=(8,8,32,32,128,128,32,32)). The only difference between 2D and 3D model was that the former used 2D convolutions and the latter used 3d convolutions.

During training we used the Adam optimizer [4]. We chose a learning rate of $1e - 3$ for the 2D models, and a learning rate of $1e - 3$ for the 3D i-RIM. The learning rate was halved every 10 epochs.

All models were trained in data parallel mode using 4 GPUs (Nvidia Tesla V100-PCIE-16GB) in Pytorch. For the RIM we chose batch size 4. This is because only a single image fits into each GPU during training. For the i-RIM 2D we used batch size 32, and for the i-RIM 3D we used batch size 4.

For evaluation we had to run the i-RIM 3D on a Nvidia Tesla V100-PCIE-32GB graphics card. Pytorch and CuDNN produce a large memory overhead when input data has varying size. Since all test volumes differ vastly in size, we always ran into out-of-memory issues with a Nvidia Tesla V100-PCIE-16GB. We will investigate this issue in the future.

## C Evaluating of Memory Consumption

We evaluated memory consumption on a single GPU in Pytorch with mini-batch size 1. In order to measure memory consumption we used the function

```
torch.cuda.max_memory_allocated()
```

After each mini-batch we reset the maximum memory allocated with

```
torch.cuda.reset_max_memory_allocated()
```

To evaluate memory consumption during training, we ran 10 mini-batches and recorded the maximum allocated memory after each back-propagation step. We report the minimum of all 10 values which excludes outliers. We use the same approach for evaluating memory consumption during test, but this time the model is set to evaluation mode, and no back-propagation is performed.