[Reviews · NeurIPS 2019]

Reviewer 1



The paper seems to propose a new (original) method, which is significant in that it enables more complex inversion problems. The paper is generally well-written (clarity), although it is quite dense for a reader not intimately familiar with the subject matter. The authors make an attempt to include information from other papers that a reader may not be familiar with. This does not always succeed: In Eq. (6), it is not clear how this layer can be trivially inverted analytically, unless function G() is trivially invertible analytically. For 2.3, it is not obvious how a layer containing of ReLUs, or downsampling at large, is invertible. This section should be expanded notably. I have a concern in quality. The paper's results validate the method on the authors' chosen task. However, the paper's underlying tone. both by the rather general title and the abstract/conclusion, is that the method is generally applicable. However, it is not clear how specialized MRI really is, or what kind of problem classes this can be applied to. To uphold the claims of generality, the authors should speak to this, or alternatively, change the title to be more specific. I personally dislike papers with very general titles that then apply only to a very narrow application area. In 4.4, it is claimed that the machine state occupies over 7% of available GPU RAM, and therefore it is not trainable with current hardware. However, there is no reason one could not page memory between GPU and CPU. Would that really make things infeasibly slow, e.g. if properly interleaved memory copies are used?

Reviewer 2



Originality This paper has an originality. The invertible recurrent inference machine (i-RIM) is a new structure. Quality The proposed method i-RIM outperforms all conventional methods. In particular, the i-RIM shows very nice results on the MRI reconstruction task. For instance, the i-RIM sits on top of the leader board of fastMRI challenge. Clarity The authors need to do a better job in their writing and motivating the problem. It is not easy to read. The text in Figure 2 is very hard to read. Please increase the font size. Significance This paper contains a new structure which shows very nice results in MRI reconstruction tasks.

Reviewer 3



This paper introduces a memory efficient method to train a model for inverse inference. This is done by introducing RIMs and use them for iterative inverse process, a series of inverse operations from the output all the way to the input. Simply, put this is basically running RIM for inverse each of the layers to bound the memory. The other part of the paper is to introduce a set of invertible layers (e.g. orthogonal convolution / residual block with spatial downsampling). These parts are somewhat apart from the proposed method above. While certainly this is a good contribution, but the coherence of the paper was blurred from adding this. My overall impression of the paper is a direct extension of the RIM paper plus a few useful tools for inversible neural network especially for vision. This makes this paper more like a swiss-army-knife than a very solid piece of in-depth work. Although I believe this paper would be very useful for Computer Vision community, somewhat blurry focus of this paper makes me hesitant to recommend this paper to be accepted as-is. -------------------------------------------------------------------------------------------------------- I thank authors' rebuttal and it did address some of my concerns (especially the coherence part). I am happy to recommend this paper to be accepted.

[Author Response · NeurIPS 2019]

We thank the reviewers for their useful feedback. Overall, we see a positive reception of our work. Reviewer 1 points out that the paper is "well-written", reviewer 4 says that "the paper has an originality" and notices the "very nice results on the MRI reconstruction task", and reviewer 5 finds that "this paper would be very useful for Computer Vision community". We start by emphasizing our main contributions to help clarify some of the issues raised by reviewers 4 & 5 about structure and focus of the paper. We have added the following statement of contributions to the Introduction:

1. **The first invertible "learning to infer" model.** We draw inspiration from generative models to modify an existing "learning to infer" approach called "Recurrent Inference Machines" to be fully invertible. This allows us to train more expressive (deeper) models than before because we can overcome memory constraints during training using invertible learning [16]. Because the presented model is invertible it will allow for semi- and un-supervised training in the future [18]. This is especially relevant for domains where the signal of interest $\mathbf{p}$ is always unknown, such as in synthesis imaging in radio astronomy.

2. **Stable invertible learning of very deep models.** To the best of our knowledge, we present the deepest network that has been successfully trained with invertible learning[1]. In practice, invertible learning can be unstable due to numerical errors that can accumulate in very deep networks [16]. In our experience, common invertible layers [18-20] suffered from this problem. We give intuitions why these layers might introduce training instabilities, and present a new layer that addresses these issues and enables stable invertible learning of very deep networks (400 layers).

3. **Scale to large observations.** We demonstrate in experiments that our model can be trained on large volumes in MRI (3d). Previous "learning to infer" models were only able to perform reconstruction on 2d slices. For data that has been acquired with 3d sequences, however, these approaches are not feasible anymore. Our approach overcomes this issue. This result is relevant for other domains with large observations such as synthesis imaging in radio astronomy.

**Reviewer 1: Generality of our approach and the question how specialized MRI really is.** Undersampled MRI image reconstruction is a difficult deconvolution problem and, as such, it is a prime example of an inverse problem. Our approach is an extension of the RIM, as recognized by reviewer 5, which has proven successful in a number of imaging tasks (l. 196-198). Here, we wanted to focus on the most challenging and practical problems where observation size is an issue. We plan to apply our approach to synthesis imaging in radio astronomy in the future, however, this application would have been beyond the scope of the presented paper. We modified the paper to emphasize the significance of accelerated MRI as an inverse problem.

**Reviewer 1: Paging memory between GPU and CPU** That is indeed infeasible. The Nvidia V100 used for training has a memory bandwidth of 900 GB/s, while the PCIe 3.0 x16 bus that connects the card to main memory has transfer rates < 16 GB/s. Data transfer between main memory and GPU memory is generally considered a bottleneck.

**Reviewer 1: Inversion of Eq. 6** For improved clarity we have changed eq. 6 to show the forward computation on the left, and the reverse computation on the right:

$$
\begin{aligned}
\mathbf{y}_1 &= \mathbf{x}_1 & \mathbf{x}_2 &= \mathbf{y}_2 - \mathcal{G}(\mathbf{y}_1) \\
\mathbf{y}_2 &= \mathbf{x}_2 + \mathcal{G}(\mathbf{y}_1) & \mathbf{x}_1 &= \mathbf{y}_1
\end{aligned}
\tag{1}
$$

We did the same for eq. 14. Please note, that the inversion holds independent on the parametrisation of $\mathcal{G}(\cdot)$, i.e. $\mathcal{G}(\cdot)$ does not have to be invertible.

**Reviewer 4: Increase font size in figure 2** Fixed.

**Reviewer 4 & 5: Reviewer 5 emphasized that the invertible layer appeared to be "somewhat apart from the proposed method", and concluding that "the coherence of the paper was blurred from adding this"** We see why the reviewer finds this unclear and we hope that the above contribution section clarifies our reasons for including the invertible layer in the paper. To elaborate more, we originally had contributions 1 and 3 as our goals. But since invertible learning for "learning to infer" models is new territory, it was not immediately clear whether established layers could guarantee stable invertible learning in this scenario. After all, it is well recognized that great care has to be taken in initialization and parametrization of Glow and real-NVP layers for stable training. In experiments, we verified that these layers indeed made training of the i-RIM unstable. Hence, we introduced a new layer that addresses the issues we identified with other layers (l.136-147), and that allowed us to train models in a stable fashion. We therefore understand our invertible layer as a crucial component to guarantee stable training of the i-RIM. To better justify the invertible layer as a component for training the i-RIM we have added more context in the introduction of section 2.2.

## Footnotes

[1] `http://bayesiandeeplearning.org/2018/papers/37.pdf` presents a 300 layer deep network trained on CIFAR10


[Meta-Review · NeurIPS 2019]

The idea of iterative inverse models with constant memory is of great interest to solve inverse problems with large number of layers. The proposed method has been successfully applied on a network with 400 layers. Even though the reviewers have raised the question of potential applicability of the technique beyond the domain of MRI reconstruction, there is clearly substantial novel material in the paper that warrants its publication as a poster. Almost all the reviewers have pointed out that the paper is densely written and needs to be revised to add more clarity.